# Factors associated with overweight and obesity among women of reproductive age in Cambodia: Analysis of Cambodia Demographic and Health Survey 2021–22

Samnang Um [1,2]*, Yom An[1,3]

**1** National Institute of Public Health, Phnom Penh, Cambodia, **2** Faculty of Social Science and Humanities, Royal University of Phnom Penh, Phnom Penh, Cambodia, **3** School of Health Sciences, Faculty of Medicine, University of the Ryukyus, Okinawa, Japan

* umsamnang56@gmail.com

**Data Availability Statement:** Our study used the 2021-2022 Cambodia Demographic and Health Survey (CDHS) datasets. The DHS data are publicly available from the website at (URL:https://www.

## Abstract

Overweight and obesity are associated with increased chronic disease and death rates globally. In Cambodia, the prevalence of overweight and obesity among women is high and increasing. This study aimed to determine the prevalence and factors associated with overweight and obesity among women of reproductive age (WRA) in Cambodia. We analyzed data from the 2021–22 Cambodia Demographic and Health Survey (CDHS). Data analysis was restricted to non-pregnant women, resulting in an analytic sample of 9,417 WRA. Multiple logistic regressions were performed using STATA V17 to examine factors associated with overweight and obesity. The prevalence of overweight and obesity among WRA was 22.56% and 5.61%, respectively. Factors independently associated with increased odds of overweight and obesity included women aged 20–29 years [AOR = 1.85; 95% CI: 1.22–2.80], 30–39 years [AOR = 3.34; 95% CI: 2.21–5.04], and 40–49 years [AOR = 5.57; 95% CI: 3.76–8.25], women from rich wealth quintile [AOR = 1.44; 95% C: 1.19–1.73], having three children or more [AOR = 1.40; 95% CI: 1.00–1.95], ever drink alcohol [AOR = 1.24; 95% CI: 1.04–1.47], and current drink alcohol [AOR = 1.2; 95% CI: 1.01–1.45]. Women completed at least secondary education were less likely being overweight and obese [AOR = 0.73; 95% CI: 0.58–0.91]. Overweight and obesity remains highly prevalent among WRA in Cambodia. Therefore, there is an urgent need to take interventions that target women from higher socio-demographic status to reduce the risk of life-threatening caused by being overweight and obese through raising awareness of important changing lifestyles.

## Introduction

Overweight and obesity are significant global public health challenges that have rapidly risen in the last four decades and are regarded as an epidemic [1]. In 2016, approximately 39% (1.9

dhsprogram.com/data/available-datasets.cfm). The Shapefiles for administrative boundaries in Cambodia are publicly accessible through the DHS website at (URL:https://spatialdata.dhsprogram.com/boundaries/#view=table&countryId=KH).

**Funding:** The authors received no specific funding for this work.

**Competing interests:** The authors have declared that no competing interests exist.

billion) of adults aged 18 years and older were overweight, and 13% (650 million) were obese worldwide [1]. Being overweight and obese are major risk factors for several non-communicable diseases (NCDs), including cardiovascular (CVDs), kidney diseases, type 2 diabetes, some cancers, musculoskeletal disorders, and other chronic diseases [2, 3]. Moreover, among women of reproductive age (WRA), overweight and obesity have been associated with increased risk of pregnancy complications, cesarean section births, adverse birth outcomes, and infant mortality [4]. Overweight and obesity are the fourth leading cause of risk-attributable mortality [5], with a reduced life expectancy of 5–20 years, depending on the condition's severity and comorbidities [1]. According to the WHO, being overweight and obese are the leading risks for global deaths, with at least 2.8 million adults dying annually due to these conditions [1]. Overweight and obesity were higher among women than men in both developed and developing countries. Getahun H. et al. found that overweight was 40% in women vs. 39% in men, and obesity was 15% in women vs. 11% in men [1]. The Global Nutrition Report 2019 showed that 26.1% of women and 20.4% of men were overweight, while obesity was 6.3% in women compared to 3.5% in men [6]. Along with the rapidly increasing population in Cambodia, from 15.42 million in 2015 to nearly 16.21 million in 2019 [7], the prevalence of overweight and obesity is also steadily rising. According to the Cambodia STEPS survey, the overall prevalence of current smoking any tobacco was 29.4% in 2010 to 22.1% in 2016, and current drinking alcohol was 53.5% in 2010 to 46% in 2016, 11% were low physical activity in 2010 to 13% in 2016 [11]. In addition, 63% of women consumed sweet beverages, and 33% consumed unhealthy foods the previous day in 2021–22 [8]. The 2021–22 Cambodia Demographic and Health Survey (CDHS) report indicated that overweight and obesity among non-pregnant women of reproductive age increased from 18% (15.2% overweight and 2.8% obese) in 2014 to 33% (26% overweight and 6% obese) in 2021–22 [8]. It was estimated that overweight and obese contribute to rising healthcare costs in Cambodia (approximately 1.7% of its annual gross domestic product (GDP) per capita) and is a significant contributor to mortality and decreased general health and productivity [9]. Predictors of overweight and obesity among WRA from Demographic and Health Survey (DHS) data such as Cambodia, Bangladesh, Nepal, India, and Ethiopia include higher socioeconomic status, older age, marriage, living in an urban residence, and lack of education [3, 10–12]. Women with formal employment had higher odds of being overweight or obese than informally employed women [3, 13]. Being overweight and obese were more common in women who used hormonal contraceptives such as oral contraceptive pills, implants, patches, and rings [14, 15]. Globally, 30% of daily smokers are overweight or obese [16], with women smokers at greater risk for obesity than men smokers [17, 18]. Women with frequent television watching [19], alcohol drinking [20, 21], and regular consumption of sweets foods and unhealthy foods [22] were found to have higher odds of being overweight or obese. To our knowledge, factors associated with overweight and obesity, specifically among WRA in Cambodia using updated data, have not been explored. A previous study on the prevalence of overweight and obesity among WRA and its associated factors utilized data since 2014 [3]. In Cambodia, where the prevalence of overweight and obesity in WRA has increased rapidly, a comprehensive investigation of a wide range of socio-demographic and behavioral factors is warranted to identify the factors independently associated with having overweight and obesity. Identifying the critical modifiable socio-demographic and behavioral factors, as well as women at a high risk of being overweight and obese, may help guide the timely development of promising and feasible public health intervention strategies to address the growing overweight and obesity pandemic. Therefore, we aimed to determine the prevalence and examined socio-demographic and behavioral factors associated with overweight and obesity among WRA in Cambodia.

## Material and methods

### Ethics statement

The study data used in this study were women's data, which were extracted from the most recent CDHS 2021–22 [8], which are publicly available with all personal identifiers of study participants removed. Also, the CDHS data are publicly accessible and were made available to us upon request through the DHS website at (URL: https://dhsprogram.com/data/available-datasets.cfm). Written informed consent was obtained from the parent/guardian of each participant under 18. The data collection tools and procedures for CHDS 2021–22 were approved by the Cambodia National Ethics Committee for Health Research on 10 May 2021 (Ref: 083 NECHR) and the Institutional Review Board (IRB) of ICF in Rockville, Maryland, USA.

### Data sources and procedures

We followed the methods of Um S et al., 2023 [3]. To analyze the prevalence and factors associated with overweight and obesity among WRA in Cambodia, we used existing women's data from the 2021–22 CDHS, which was a nationally representative population-based household survey implemented by the National Institute of Statistics (NIS) in collaboration with the Ministry of Health (MoH). Data was collected from September 15, 2021, to February 15, 2022 [8]. The sampling frame used for the 2021–22 CDHS was taken from the Cambodia General Population Census 2019 [7]. After that, the participants were selected using probability proportion based on two-stage stratified cluster sampling from the chosen sampling frame. In the initial stage, 709 enumeration areas (EAs) (241 urban areas and 468 rural areas) were selected. In the second stage, an equal systematic sample of 25–30 households was selected from each cluster of 21,270 households. In total, 19,496 women aged 15–49 were interviewed, with a response rate of 98.2%. Survey data were obtained through face-to-face interviews using a standardized survey instrument by trained interviewers. Anthropometric weight and height measurements for adult women aged 15–49 were taken by trained female field staff using standardized instruments and procedures [7]. Weight measurements were taken using scales with a digital display (UNICEF model S0141025). Height was measured using a portable adult height measurement system (UNICEF model S0114540). The detailed protocol and methods were published previously [8]. Eligible participants for this study were women of reproductive age 15–49 years, with the exclusion of pregnant women and those who had a birth within two months before the survey, with available body mass index (BMI) data. As a result, we excluded 828 pregnant women and 9,249 women with missing BMI data. A final sample included in this analysis was 9,417 women of reproductive age 15–49 years.

### Measurements

#### Outcome variable

The primary outcome variable of this study was overweight and obese for women of reproductive age. BMI—defined by dividing a person's weight in kilograms by the square of their height in meters ($kg/m^2$)–was used to measure the outcome [8, 23]. For adults over 15 years old, the following BMI ranges were used: Underweight ($\leq 18.4 \, kg/m^2$), Normal weight ($18.5–24.9 \, kg/m^2$), Overweight ($25.0–29.9 \, kg/m^2$), and Obesity ($\geq 30.0 \, kg/m^2$) [8, 23]. **Overweight/Obese** was defined as a binary outcome for which women with a BMI $\geq 25.0 \, kg/m^2$ were classified as overweight and obese (coded = 1), while women with a BMI $< 25.0 \, kg/m^2$ were coded as other (coded = 0).

### Independent variables

Independent variables consisted of sociodemographic characteristics and behavioral factors, women's age in years (15–19, 20–29, 30–39, and 40–49), marital status (not married, married or living together, and divorced or widowed or separated), educational level (no formal education, primary, secondary or higher education), occupation (not working, agriculture, manual labor or unskilled, professional or sealer or services), and number of children ever born (no children, one-two child, and three and more children). Households' wealth status was represented by a wealth index calculated via principal component analysis (PCA) and using variables for household assets and dwelling characteristics. Weighted scores are divided into five wealth quintiles (poorest, poorer, medium, richer, and richest), each comprising 20% of the population [8] and place of residence (rural vs. urban). Cambodia's provinces were regrouped for analytic purposes into a categorical variable with four geographical regions: plains, Tonle Sap, coastal/sea, and mountains [7]. Behavioral factors included smoking (non-smoker vs. smoker), alcohol consumption in the past month (never drink, ever drink, and current drink), current alcohol drinking corresponded to one can or bottle of beer, one glass of wine, or one shot of spirits in the past month, watching television at least once a week (yes vs. no), contraceptive usage (not used, hormonal methods (using a pill, emergency contraceptive pill, Norplant, injection, and vaginal rings), non-hormonal methods (condoms, the diaphragm, the IUD, spermicides, lactational amenorrhea, and sterilization), and traditional methods (periodic abstinence and withdrawal) [8].

### Statistical analysis

Statistical analyses were performed using STATA version 17 (Stata Corp 2021, College Station, TX) [24]. We accounted for CDHS sampling weight and complex survey design using the survey package in our descriptive and logistic regression analyses. Key socio-demographic characteristics and behavioral factors were described in weighted frequency and percentage. The provincial variation in the prevalence of overweight and obesity was done using ArcGIS software version 10.8 [24]. A shapefile for Cambodian administrative boundaries was obtained from the United Nations for Coordination of Humanitarian Affairs (OCHA) https://data.humdata.org/dataset/cod-ab-khm?. License (https://data.humdata.org/faqs/licenses).

Bivariate analysis using Chi-square tests was used to assess associations between independent variables (socio-demographic characteristics and behavioral factors) and **Overweight/Obese**. Variables associated with overweight and obese at p-value $\leq 0.10$ [3] or that had a potential confounder variable (for example, women's age, household wealth index, alcohol consumption, and place of residence) were included in the final multiple logistic regression analyses. Simple logistic regression was used to determine the magnitude effect of associations between overweight and obesity with socio-demographic characteristics and behavioral factors. Results are reported as odds ratios (OR) with 95% confidence intervals (CI). Multiple logistics regression was then used to assess independent factors associated with overweight and obesity after adjusting for other potential confounding factors in the model. Results from the final adjusted model are reported as adjusted odds ratios (AOR) with 95% CI and corresponding p-values. Multicollinearity between independent variables was checked before fitting the final regression model, including women's age, number of children ever born, education, household wealth index, occupation, marital status, place of residence, and geographical regions. The result of the evaluation of variance inflation factor (VIF) scores after fitting an Ordinary Least Squares regression model with the mean = 1.44 indicated no collinearity concerns [25] (see **S1 Table**).

## Results

### Characteristics of the study samples

The mean age of women was 31 years old (SD = 9.6 years), of which the 30–49 age group accounted for 57.11%. Almost 67.82% were married, 49.47% had completed at least secondary education, and 11.90% did not receive formal schooling. 30.30% of women did professional work, and 25.86% were unemployed. Of the total sample, 35.97% of women were from poor households. Over half (57.27%) of women resided in rural areas. About 27.79% had three or more children, while 29.49% had no children. Only 1.40% of women reported cigarette smoking, 16.60% reported currently drinking alcohol, 18.6% reported ever drinking, and 12.35% reported using hormonal contraceptives. The mean women's BMI was 22.9 kg/m$^2$ (SD = 3.9 kg/m$^2$), and 22.56% and 5.61% were overweight and obese, respectively (see **Table 1**).

### Distribution of overweight and obesity among WRA by provinces

The prevalence of overweight and obesity is highest among WRA in Kampong Cham (34.1%), Kandal (32.3%), Svay Rieng (32%), Preah Sihanouk (31.2%), Phnom Penh, Kompong Thom, Pailin, and Tboung Khmum (30% each) and lowest among WRA in Ratanak Kiri (9.4%) and Kampong Chhnang (14.6%) (see **Fig 1**, and **S2 Table**).

### Factors associated with overweight and obesity in bivariate analysis

In bivariate analysis, all socio-demographic characteristics and behavioral factors were significantly associated with overweight and obesity, except smoking status and frequency of watching television (**Table 2**). Women had a higher prevalence of overweight/obesity if they were aged 40–49 years (47.31%) compared to younger age groups (p-value <0.001), married (35.87%) compared to another relationship status (p-value <0.001), had no formal education (37.91%) compared to higher education (p-value <0.001), were employed in a professional job (34.03%) compared to other careers, or were from the rich households (30.90%) compared to poor households (p-value <0.001). In addition, Overweight/obesity increased with parity. Women reported at least three children had a significantly higher proportion of overweight and obesity (42.53%) compared to those who had children less than three (p-value < 0.001). Women who currently drink alcohol had a significantly higher proportion of overweight and obese (33.87%) than women who did not (p-value <0.001). Women who used hormonal contraceptive methods had a significantly higher proportion of overweight and obesity (37.92%) than women who did not use contraceptives (p-value <0.001). Geographic regions of residence were likewise associated with a woman being overweight and obese. Women who lived in urban areas had a higher prevalence of overweight and obesity than those living in rural areas (29.90% vs 26.87%, p-value = 0.028). Plain regions were positively associated with overweight and obesity (29.91%) compared to the other areas (p-value = 0.006).

### Factors associated with overweight and obesity in adjusted logistic regression

As shown in **Table 3**, several factors were independently associated with increased odds of being overweight and obese among women. These factors included age group 20–29 years [AOR = 1.85; 95% CI: 1.22–2.80], 30–39 years [AOR = 3.34; 95% CI: 2.21–5.04], and 40–49 years [AOR = 5.57; 95% CI: 3.76–8.25] married [AOR = 2.49; 95% CI: 1.71–3.62] and widowed/divorced/separated [AOR = 1.14; 95% CI: 1.14–2.63], middle wealth quintile [AOR = 1.21; 95% CI: 1.02–1.44], and rich wealth quintile [AOR = 1.44; 95% CI: 1.19–1.73], having at least three children or more [AOR = 1.40; 95% CI: 1.00–1.95], ever drink alcohol

**Table 1. Socio-demographic and behavior characteristics of the weighted samples of women aged 15–49 years old in Cambodia, 2021–2022 (n = 9,417, weighted count).**

| Variables | | Freq. | Percent (95% CI) |
|---|---|---|---|
| **Women's mean age in years (SD)** | | | **30.9 (9.6)** |
| | 15–19 | 1,488 | 15.80 |
| | 20–29 | 2,551 | 27.09 |
| | 30–39 | 3,220 | 34.19 |
| | 40–49 | 2,158 | 22.92 |
| **Marital status** | | | |
| | Not married | 2,429 | 25.79 |
| | Married or living together | 6,387 | 67.82 |
| | Widowed/divorced/separated | 601 | 6.38 |
| **Education** | | | |
| | No education | 1,121 | 11.90 |
| | Primary | 3,637 | 38.62 |
| | Secondary and higher | 4,659 | 49.47 |
| **Current work status** | | | |
| | Not working | 2,435 | 25.86 |
| | Agricultural | 1,607 | 17.06 |
| | Professional | 2,853 | 30.30 |
| | Manual labor and unskilled | 2,331 | 24.75 |
| **Number of children born** | | | |
| | No birth | 2,777 | 29.49 |
| | 1–2 child | 4,023 | 42.72 |
| | Three or above | 2,617 | 27.79 |
| **Household wealth index** | | | |
| | Poor | 3,387 | 35.97 |
| | Middle | 1,822 | 19.35 |
| | Rich | 4,207 | 44.67 |
| **Smoking** | | | |
| | Non-smoker | 9,285 | 98.60 |
| | Smoker | 132 | 1.40 |
| **Current drinking alcohol** | | | |
| | Never drink | 6103 | 64.80 |
| | Ever drink | 1751 | 18.60 |
| | Current drink | 1562 | 16.60 |
| **Frequency of watching television** | | | |
| | Not at all | 5,808 | 61.68 |
| | Less than once a week | 1,493 | 15.85 |
| | At least once a week | 2,116 | 22.47 |
| **Ever report of contraceptive use** | | | |
| | No method | 5,152 | 54.71 |
| | Traditional method | 2,398 | 25.46 |
| | Non-hormonal method | 704 | 7.48 |
| | Hormonal method | 1,163 | 12.35 |
| **Place of residence** | | | |
| | Urban | 4,024 | 42.73 |
| | Rural | 5,393 | 57.27 |
| **Region** | | | |

*(Continued)*

**Table 1.** (Continued)

| Variables | | Freq. | Percent (95% CI) |
|---|---|---|---|
| | Plain | 4,708 | 49.99 |
| | Tonle Sap | 2,859 | 30.36 |
| | Coastal | 595 | 6.32 |
| | Plateau/Mountain | 1,255 | 13.33 |
| BMI means in kg/m2 (SD) | | | 22.9 (3.9) |
| | Underweight | 1013 | 10.76 (9.9–11.7) |
| | Normal weight | 5751 | 61.08 (59.7–62.5) |
| | Overweight | 2124 | 22.56 (21.5–23.7) |
| | Obese | 528.2 | 5.61 (5.0–6.3) |

**Notes:** Survey weights are applied to obtain weighted percentages. **Plains:** Phnom Penh, Kampong Cham, Tbong Khmum, Kandal, Prey Veng, Svay Rieng, and Takeo; **Tonle Sap:** Banteay Meanchey, Kampong Chhnang, Kampong Thom, Pursat, Siem Reap, Battambang, Pailin, and Otdar Meanchey; **Coastal/sea:** Kampot, Kep, Preah Sihanouk, and Koh Kong; **Mountains:** Kampong Speu, Kratie, Preah Vihear, Stung Treng, Mondul Kiri, and Ratanak Kiri.

[AOR = 1.24; 95% CI: 1.04–1.47], and current drink alcohol [AOR = 1.21; 95% CI: 1.01–1.45]. On the contrary, the following factors were independently associated with decreased odds of being overweight and obese: women with at least secondary education [AOR = 0.73; 95% CI: 0.58–0.91], working in manual labor jobs [AOR = 0.76; 95% CI: (0.64–0.90] (see **Fig 2**).

## Discussion

The overall prevalence of overweight and obesity among WRA in Cambodia was 22.56% (95% CI: 21.5–23.7) and 5.61% (95% CI: 5.0–6.3), which was higher than the percentage of WRA who were the overweight and obesity in 2000 (6% and 2%), as well as in 2014 (18% and 2.8%) [3]. This continuous increase is becoming a public health problem in the country. Our findings were consistent with the findings in Global Nutrition Report 2019 where 26.1% and 6.3% of women were overweight and obese respectively [6]. At the same time, overweight and obesity is highest proportion among WRA residing in active economic provinces such as Kampong Cham, Kandal, Svay Rieng, Preah Sihanouk, Phnom Penh (see **Fig 1**). The variation and increase in overweight and obesity in Cambodia could be due to lifestyle changes and urbanization [11]. For example, the consumption of fast foods high in sugar, fat and salt has become widespread [11]. High prevalence of women consumed sweet beverages, and consumed unhealthy foods the previous day and drinking alcohol in past 30 days were also fund in CDHS 2021–22 [8]. Furthermore, physical activity has decreased due to improved transportation [8, 11].

Older aged women were more likely to be overweight or obese as compared to younger women aged 15–19 years. This is consistent with other studies that showed obesity was more prevalent in older WRA [3, 26]. The risk of women becoming overweight or obese rises with age, possibly due to unhealthy food consumption and a lack of physical activity [27]. Married women were more likely to be overweight or obese as compared to single women. Similar to previous study on the prevalence of overweight and obesity among WRA and its associated factors utilized CDHS data since 2014 [3], and in line with pooled analysis of DHS data among WRA in Bangladesh, Nepal, and India [28]. The explanation for that could be that married women busy working and take care their children and family may they do not pay more attention to nutritional and physical activity [3]. Women with better socioeconomic status and who live in urban areas had higher odds of being overweight and obese, which is consistent with

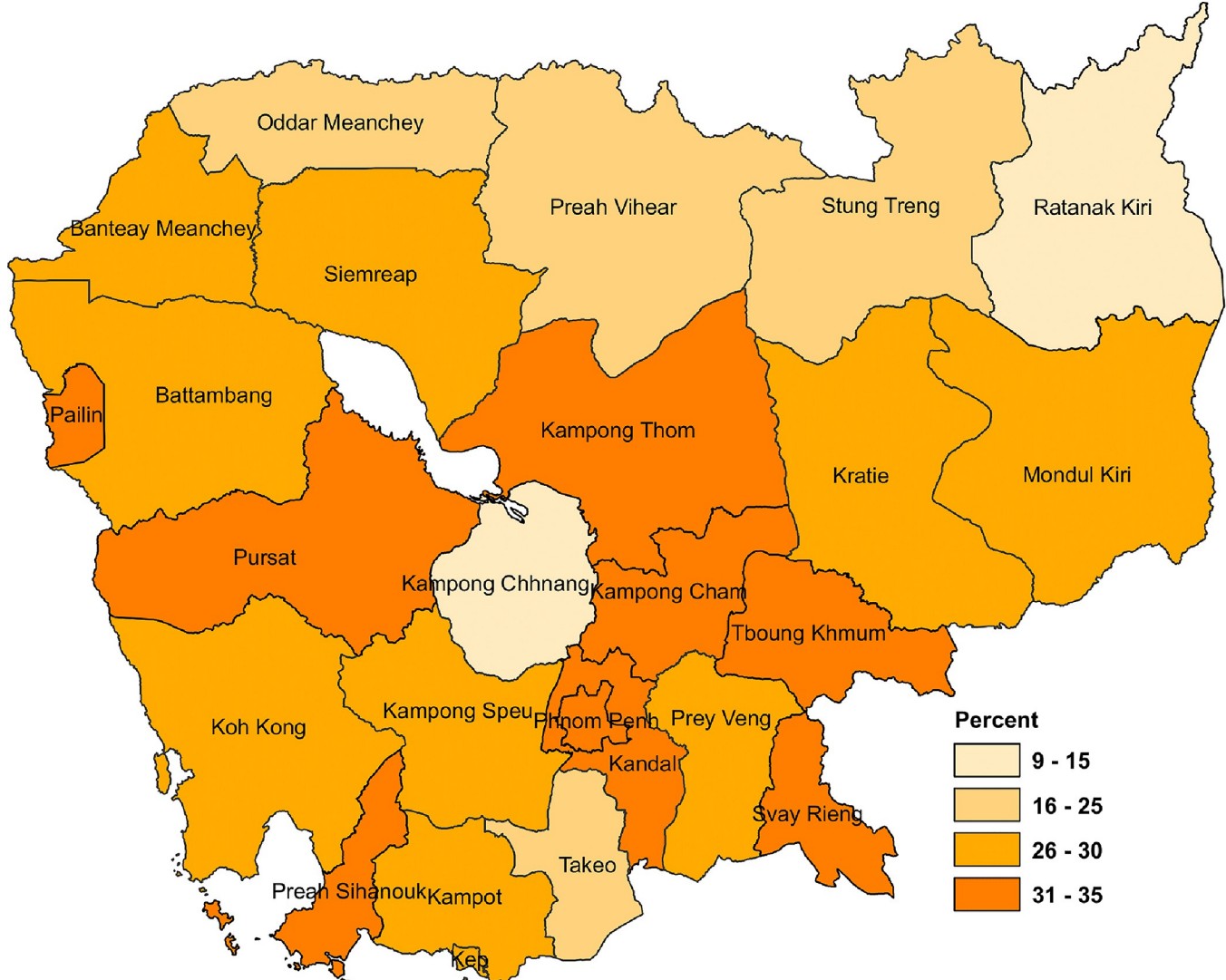

**Fig 1. Prevalence of overweight and obesity in Women Reproductive Age by province.** The map was created using ArcGIS software version 10.8 [24]. A shapefile for Cambodian administrative boundaries was obtained from the United Nations for Coordination of Humanitarian Affairs (OCHA) https://data.humdata.org/dataset/cod-ab-khm?. License (https://data.humdata.org/faqs/licenses).

prior research [29, 30]. Women with higher socioeconomic status tend to use more improved technologies for a more comfortable lifestyle [19, 31]. Women have at least three and more children at risk of being overweight and obese, similar to other studies [3, 32]. During pregnancy, factors such as stress, depression, and anxiety may play a role in hypothalamic-pituitary-adrenal hyperactivity [33, 34]. Women with several children may also have gained weight due to their reduced physical activity and have less time to focus on health behaviors, including weight management. In addition, women ever and currently consuming alcohol were significantly associated with higher risks of being overweight and obese. The association between alcohol consumption status and overweight and obesity has been studied thoroughly in various populations, such as Urban Cambodia and Hawassa City, southern Ethiopia [20, 21]. Commonly, alcohol consumption is considered to increase appetite, resulting in excessive energy intake and leading to overweight and obesity [35]. In contrast, women with higher

**Table 2.** Factors associated with overweight and obesity among women aged 15–49 years in Chi² analysis (n = 9,417, weighted count).

| Characteristics | | Overweight/Obese (n = 2,652) | | Normal/Underweight (n = 6,764) | | P value |
|---|---|---|---|---|---|---|
| | | Freq. | % | Freq. | % | |
| Women's age group in years | | | | | | |
| | 15–19 | 80 | 5.38 | 1,408 | 94.62 | <0.001 |
| | 20–29 | 444 | 17.40 | 2,107 | 82.60 | |
| | 30–39 | 1,108 | 34.41 | 2,112 | 65.59 | |
| | 40–49 | 1,021 | 47.31 | 1,137 | 52.69 | |
| Marital status | | | | | | |
| | Not married | 181 | 7.45 | 2,248 | 92.55 | <0.001 |
| | Married or living together | 2,291 | 35.87 | 4,096 | 64.13 | |
| | Widowed/divorced/separated | 181 | 30.12 | 420 | 69.88 | |
| Education | | | | | | |
| | No formal education | 425 | 37.91 | 695 | 62.00 | <0.001 |
| | Primary | 1,254 | 34.48 | 2,383 | 65.52 | |
| | Secondary and higher | 973 | 20.88 | 3,686 | 79.12 | |
| Current work status | | | | | | |
| | Not working | 586 | 24.07 | 1,849 | 75.93 | <0.001 |
| | Agricultural | 485 | 30.18 | 1,121 | 69.76 | |
| | Professional | 971 | 34.03 | 1,881 | 65.93 | |
| | Manual labor and unskilled | 563 | 24.15 | 1,768 | 75.85 | |
| Number of children born | | | | | | |
| | No birth | 264 | 9.51 | 2,512 | 90.46 | <0.001 |
| | 1–2 child | 1,275 | 31.69 | 2,748 | 68.31 | |
| | Three or above | 1,113 | 42.53 | 1,504 | 57.47 | |
| Household wealth index | | | | | | |
| | Poor | 841 | 24.83 | 2,546 | 75.17 | <0.001 |
| | Middle | 512 | 28.10 | 1,311 | 71.95 | |
| | Rich | 1,300 | 30.90 | 2,907 | 69.10 | |
| Smoking | | | | | | |
| | Non-smoker | 2,621 | 28.23 | 6,664 | 71.77 | 0.325 |
| | Smoker | 32 | 24.24 | 100 | 75.76 | |
| Current drinking alcohol | | | | | | |
| | Never drink | 1,573 | 25.77 | 4,530 | 74.23 | <0.001 |
| | Ever drink | 550 | 31.41 | 1,201 | 68.59 | |
| | Current drink | 529 | 33.87 | 1,033 | 66.13 | |
| Frequency of watching television | | | | | | |
| | Not at all | 1,623 | 27.94 | 4,186 | 72.07 | 0.901 |
| | Less than once a week | 423 | 28.33 | 1,069 | 71.60 | |
| | At least once a week | 606 | 28.64 | 1,509 | 71.31 | |
| Ever report of contraceptive use | | | | | | |
| | No method | 1,132 | 21.97 | 4,020 | 78.03 | <0.001 |
| | Traditional method | 821 | 34.24 | 1,577 | 65.76 | |
| | Non-hormonal method | 259 | 36.79 | 446 | 63.35 | |
| | Hormonal method | 441 | 37.92 | 722 | 62.08 | |
| Place of residence | | | | | | |
| | Urban | 1,203 | 29.90 | 2,821 | 70.10 | 0.028 |
| | Rural | 1,449 | 26.87 | 3,944 | 73.13 | |

(*Continued*)

**Table 2.** (Continued)

| Characteristics | | Overweight/Obese (n = 2,652) | | Normal/Underweight (n = 6,764) | | P value |
|---|---|---|---|---|---|---|
| | | Freq. | % | Freq. | % | |
| Region | | | | | | |
| | Plain | 1,408 | 29.91 | 3,299 | 70.07 | **0.006** |
| | Tonle Sap | 768 | 26.86 | 2,092 | 73.17 | |
| | Coastal | 171 | 28.74 | 425 | 71.43 | |
| | Plateau/Mountain | 306 | 24.38 | 949 | 75.62 | |

Notes: Survey weights are applied to obtain weighted percentages. **Plains:** Phnom Penh, Kampong Cham, Tbong Khmum, Kandal, Prey Veng, Svay Rieng, and Takeo; **Tonle Sap:** Banteay Meanchey, Kampong Chhnang, Kampong Thom, Pursat, Siem Reap, Battambang, Pailin, and Otdar Meanchey; **Coastal/sea:** Kampot, Kep, Preah Sihanouk, and Koh Kong; **Mountains:** Kampong Speu, Kratie, Preah Vihear, Stung Treng, Mondul Kiri, and Ratanak Kiri.

**Table 3. Risk factors associated with overweight and obesity in unadjusted and adjusted logistic regression analysis.**

| Characteristics | | Unadjusted (n = 9,417) | | Adjusted (n = 9,225) | |
|---|---|---|---|---|---|
| | | OR | 95%CI | AOR | 95%CI |
| Women's age group in years | | | | | |
| | 15–19 | Ref. | | Ref. | |
| | 20–29 | 3.71*** | (2.61–5.27) | **1.85*** | **(1.22–2.80)** |
| | 30–39 | 9.25*** | (6.70–12.77) | **3.34*** | **(2.21–5.04)** |
| | 40–49 | 15.84*** | (11.68–21.47) | **5.57*** | **(3.76–8.25)** |
| Marital status | | | | | |
| | Not married | Ref. | | Ref. | |
| | Married or living together | 6.96*** | (5.67–8.54) | **2.49*** | **(1.71–3.62)** |
| | Widowed/divorced/separated | 5.34*** | (3.81–7.50) | **1.73** | **(1.14–2.63)** |
| Education | | | | | |
| | No education | Ref. | | Ref. | |
| | Primary | 0.86 | (0.72–1.03) | 0.97 | (0.80–1.17) |
| | Secondary and higher | 0.43*** | (0.35–0.53) | **0.73*** | **(0.58–0.91)** |
| Current work status | | | | | |
| | Not working | Ref. | | Ref. | |
| | Agricultural | 1.37*** | (1.14–1.64) | 0.85 | (0.69–1.03) |
| | Professional | 1.63*** | (1.40–1.89) | 1.13 | (0.93–1.35) |
| | Manual labor and unskilled | 1.00 | (0.85–1.19) | **0.76*** | **(0.64–0.90)** |
| Number of children born | | | | | |
| | No birth | Ref. | | Ref. | |
| | 1–2 child | 4.41*** | (3.61–5.38) | 1.23 | (0.89–1.68) |
| | Three and above | 7.03*** | (5.79–8.55) | **1.40** | **(1.00–1.95)** |
| Household wealth index | | | | | |
| | Poor | Ref. | | Ref. | |
| | Middle | 1.18** | (1.02–1.37) | **1.21** | **(1.02–1.44)** |
| | Rich | 1.35*** | (1.18–1.56) | **1.44*** | **(1.19–1.73)** |
| Smoking | | | | | |
| | Non-smoker | Ref. | | - | - |
| | Smoker | 0.81 | (0.52–1.24) | - | - |
| Current drinking alcohol | | | | | |
| | Never drink | Ref. | | Ref. | |

*(Continued)*

**Table 3.** (Continued)

| Characteristics | | Unadjusted (n = 9,417) | | Adjusted (n = 9,225) | |
|---|---|---|---|---|---|
| | | OR | 95%CI | AOR | 95%CI |
| | Ever drink | 1.32*** | (1.13–1.54) | **1.24**** | **(1.04–1.47)** |
| | Current drink | 1.47*** | (1.23–1.76) | **1.21**** | **(1.01–1.45)** |
| **Watching television** | | | | | |
| | Not at all | Ref. | | - | - |
| | Less than once a week | 1.02 | (0.85–1.23) | - | - |
| | At least once a week | 1.04 | (0.89–1.21) | - | - |
| **Ever report of contraceptive use** | | | | | |
| | No method | Ref. | | Ref. | |
| | Traditional method | 1.85*** | (1.61–2.12) | 1.02 | (0.86–1.21) |
| | Non-hormonal method | 2.06*** | (1.64–2.60) | 0.84 | (0.65–1.10) |
| | Hormonal method | 2.17*** | (1.75–2.69) | 1.01 | (0.81–1.26) |
| **Place of residence** | | | | | |
| | Rural | Ref. | | Ref. | |
| | Urban | 1.16** | (1.02–1.33) | 1.11 | (0.94–1.31) |
| **Region** | | | | | |
| | Plain | Ref. | | Ref. | |
| | Tonle Sap | 1.33*** | (1.11–1.58) | 1.17 | (0.95–1.43) |
| | Coastal | 1.14 | (0.95–1.36) | 1.11 | (0.91–1.35) |
| | Plateau/Mountain | 1.25** | (1.00–1.55) | 1.10 | (0.86–1.41) |

**Notes: Survey weights are applied to obtain weighted percentages.** *** $p<0.01$

** $p<0.05$

* $p<0.10$.

education are less likely to be overweight or obese than women with no school due to increased knowledge and awareness; for example, studies in Cambodia, Nigeria, and South Korea reported that educated women had a lower risk of being overweight or obese and that this may be linked to education influencing healthy behavior [3, 36–39]. Education is a critical predictor of women's healthy behaviors and health outcomes, including diet and physical activity [40]. Although overweight and obesity were not significantly associated with geographical regions of residence and place of residence in multiple logistic regression, we found that the rates of overweight and obesity among women in the Plain and Coastal regions were higher than those living in other regions, at 29.91% and 28.74%, respectively. Also, the prevalence of overweight and obesity among women in urban regions was greater than that of women in rural areas (29.90% vs. 26.87). Previous research on overweight and obesity among women of reproductive age in Cambodia found that women residing in urban had a significant association with overweight and obesity [3], suggesting a need for further regionally representative studies.

Our study had few limitations. First, because CDHS 2021–2022 collected cross-sectional data, our analysis could not explore changes over time. Second, data on several critical variables, such as women's food-consumption behaviors, were not collected by CDHS, and CDHS did not objectively measure physical activity. Therefore, our ability to examine the association of these variables with overweight and obesity was restricted. Third, numerous psychological factors (e.g., depressive and anxiety disorders) and physiological factors may also be associated with overweight and obesity. Still, these factors were not included in this study because they were unavailable in the data set. Finally, because the data were available only for women aged

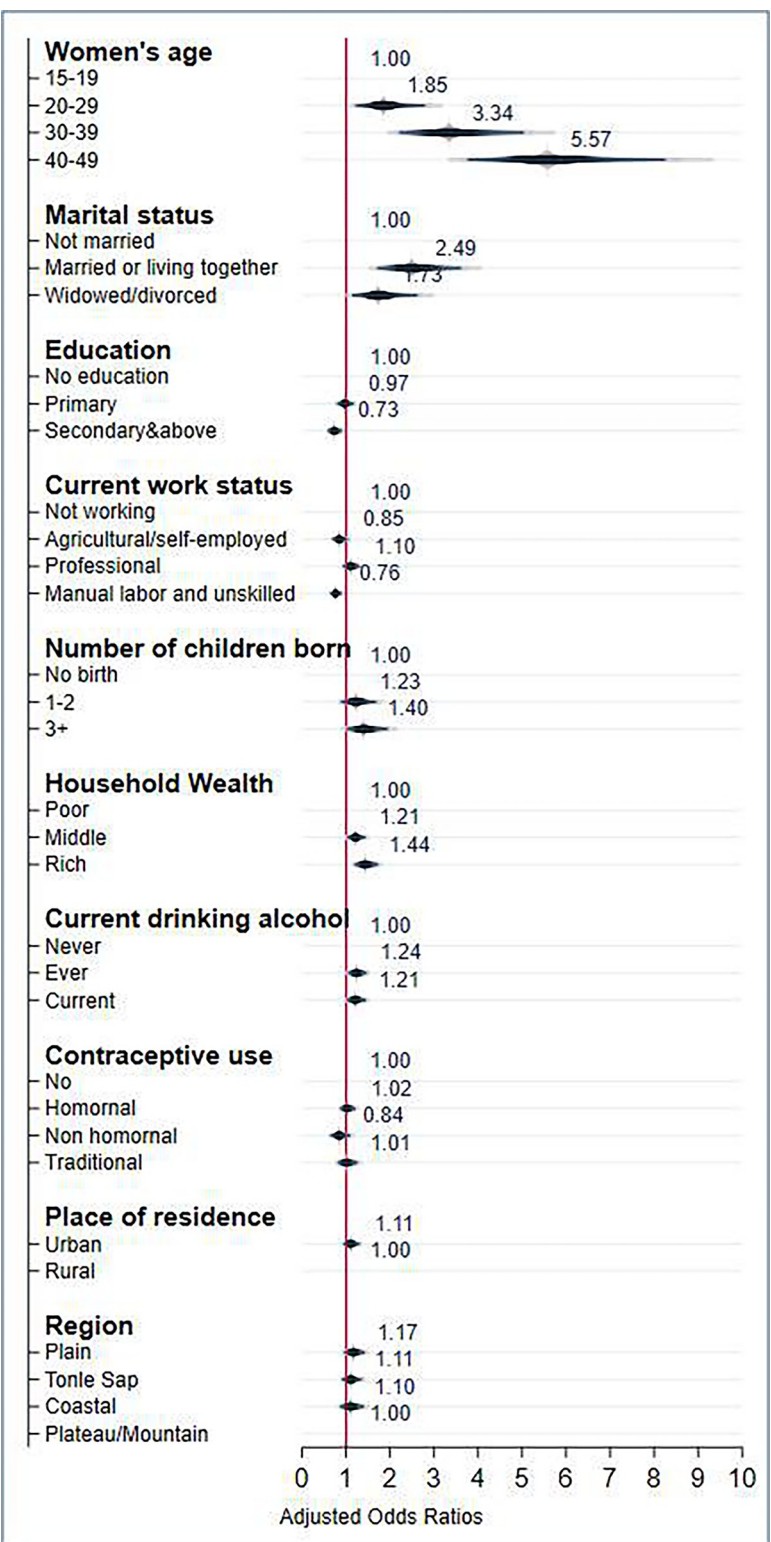

**Fig 2. Adjusted analyses of the odds of women of reproductive age being overweight and obese in Cambodia (n = 9,225).**

15 to 49, our results could not be generalized to girls younger than 15 and women older than 49. Despite these limitations, our results contributed to the literature on the prevalence and association between socio-demographic and behavioral variables and the status of overweight and obesity among women of reproductive age in Cambodia. However, the major strength of this study was the use of nationally representative data with a high response rate of 97% to examine the prevalence and predictors of overweight and obesity in this country. Data were collected using validated survey methods, including calibrated measurement tools and highly trained data collectors, contributing to improved data quality [41]. Finally, incorporating the complex survey design and sampling weights into the analysis bolstered the rigor of the findings and enables generalizing our findings to the population of non-pregnant women in Cambodia.

In conclusion, the present study indicated that the prevalence of overweight and obesity among WRA in Cambodia was very high during 2021–22. Furthermore, the main factors that significantly increased the odds of being overweight and obese were older age, married status, living in a rich household wealth index, and drinking alcohol, while higher education levels and manual labor or unskilled employment reduced the odds of being overweight and obesity. Therefore, there is an urgent need to take interventions that target women from higher socio-demographic status to reduce the risk of life-threatening caused by being overweight and obese through raising awareness of the importance of consuming healthy food, and the benefits of regular physical activity, especially among older women monitor their weight, blood pressure and blood sugar with healthcare profession.

## Supporting information

**S1 Table. Results of checking multicollinearity using Variance Inflation Factor (VIF).**
(DOCX)

**S2 Table. Prevalence of overweight and obesity among women reproductive age, CDHS 2021–2022 (n = 9,417).**
(DOCX)

## Acknowledgments

The authors would like to thank DHS-ICF, who approved the data used for this paper.

## Author Contributions

**Conceptualization:** Samnang Um.

**Data curation:** Samnang Um.

**Formal analysis:** Samnang Um.

**Funding acquisition:** Samnang Um.

**Investigation:** Samnang Um.

**Methodology:** Samnang Um.

**Project administration:** Samnang Um.

**Resources:** Samnang Um.

**Software:** Samnang Um.

**Supervision:** Samnang Um, Yom An.

**Validation:** Samnang Um.

**Visualization:** Samnang Um.

**Writing – original draft:** Samnang Um.

**Writing – review & editing:** Samnang Um, Yom An.

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
