## [Decision Letter · Decision Letter 0]

4 Oct 2023

PGPH-D-23-01290

Factors Associated with Overweight and Obesity among Women of Reproductive Age in Cambodia: Analysis of Cambodia Demographic and Health Survey

Dear Dr. Um,

Thank you for submitting your manuscript to PLOS Global Public Health. After careful consideration, we feel that it has merit but does not fully meet PLOS Global Public Health’s publication criteria as it currently stands. Therefore, we invite you to submit a revised version of the manuscript that addresses the points raised during the review process.

The review's comment about a Directed Acyclic Graph (DAG) seems like something that could be a valuable addition to your exposition, and I would like to see your revision include a DAG that represents the causal relationships that you hypothesize are present between the factors you have included in your analysis.  I believe that this DAG will also provide a crisp framework for you to address the reviewer's question about what specifically your paper adds to the existing knowledge.

We look forward to receiving your revised manuscript.

Kind regards,

Abraham D. Flaxman, Ph.D.

Academic Editor

Journal Requirements:

Additional Editor Comments (if provided):

Reviewers' comments:

Reviewer's Responses to Questions

**Comments to the Author**

1. Does this manuscript meet PLOS Global Public Health’s publication criteria? Is the manuscript technically sound, and do the data support the conclusions? The manuscript must describe methodologically and ethically rigorous research with conclusions that are appropriately drawn based on the data presented.

Reviewer #1: Partly

2. Has the statistical analysis been performed appropriately and rigorously?

Reviewer #1: No

3. Have the authors made all data underlying the findings in their manuscript fully available (please refer to the Data Availability Statement at the start of the manuscript PDF file)?

Reviewer #1: Yes

4. Is the manuscript presented in an intelligible fashion and written in standard English?

Reviewer #1: Yes

5. Review Comments to the Author

Reviewer #1: This paper aimed to determine the prevalence and factors associated with overweight and obesity among women of reproductive age (WRA) in Cambodia. The authors used the Cambodia Demographic and Health Survey (CHDS) as the data source and, therefore were able to provide nationally representative data. Considering the rise of overweight/obesity among WRA in Cambodia (i.e. 18% in 2014 to 39% in 2021-2022), this study raised an important emerging issue in the country.

Main comments:

- The first main concern is on Table 3. This analysis approach is referred as “Table 2 Fallacy”, in which all estimates were interpreted in the same way as total-effect estimates. Meanwhile, the interpretation of a confounder effect estimate may be different than for the exposure effect estimate (e.g. total effect for main exposure vs. direct effect for covariates). This is the key article explaining about this issue: https://www.ncbi.nlm.nih.gov/pmc/articles/PMC3626058/ , and here is a concrete example that quantitatively illustrate the potential of misinterpretation: https://pubmed.ncbi.nlm.nih.gov/29782045/. A Directed Acyclic Graph (DAG) that presents the conceptual framework of the study would help to carefully clarify this problem e.g. Have all potential confounders been considered? Are any of the factors on the causal pathway between one factor and the outcome? The current approach used in this paper should be acknowledged as substantial limitations to draw causal conclusions.

- Considering that the factors included in the analysis have been known associated with overweight/obesity, what does this paper add to the existing knowledge?

Introduction:

- Authors could add explanations on trends that occurred in Cambodia, e.g. lifestyle changes, that may lead to the increase of overweight/obesity prevalence among WRA. Hence, this will highlight the importance of the study.

- p.3 line 62: Why is the prevalence of overweight/obesity among WRA based on CDHS mentioned in the introduction (i.e. 39%) different from the study result (i.e. 22.56% overweight + 5.61% obesity = 28.17)?

- p.3 line 65-74: Are these predictors of overweight/obesity among WRA based on global data/other countries’ data?

Material and Methods:

Clear explanation on methods part. Below are minor feedback for this section:

- p.4 line 8-9: This “From September 15, 2021, to February 15, 2022” seems an incomplete sentence.

- p.5 line 36: “Normal weight“ should be used instead of “average weight”.

Results:

- The number of subjects included (n) can be restated at the beginning of the section.

- p.7 line 98: Missing percentage symbol on “18.60”

- p.7 line 99: Should be 22.56% instead of 22,56%

- Table 1: n of obese women had decimal i.e. 528.2 – can the author confirm this? Please check the consistency of using “()” instead of “[]” on the 95%CI in the BMI status rows.

- p.9 line 27-43 and Table 2: This paragraph was written as if the authors did post-hoc analyses, which were not the case. For instance, it could not be concluded that women who used non-hormonal contraceptive methods had a significantly higher proportion of overweight/obesity than women using other methods (line 38); This is a type of conclusion that can be drawn from a posthoc analysis or if any regrouping were done. I wonder if that is the case since the p-value for contraceptive use as the predictor on page 10 line 238 (i.e. <0.019) is different from the p-value provided in Table 2 (i.e. 0.001).

- Please add explanation for the use of bold text in certain numbers within Table 2.

Discussion:

- It needs more explanation on the findings in overweight/obesity prevalence e.g. how is this finding compared to previous studies/years and/or other countries? Why there were substantial differences in the prevalence across provinces (e.g. 34.1% in Kampong Cham vs. 9.4% in Ratanak Kiri)?

- Based on the findings of the current analysis, it needs an explanation on how marital status emerged as significant predictor of overweight/obesity.

Conclusion:

Please clarify regarding the statement “design intervention programs that target these sociodemographic factors”. For instance, given that higher education levels and income are associated with overweight/obesity in this study – should women reduce their socioeconomic status? Similarly, since older age is linked to higher odds, should interventions aiming to reducing women’s age?

6. PLOS authors have the option to publish the peer review history of their article (what does this mean?). If published, this will include your full peer review and any attached files.

**Do you want your identity to be public for this peer review?** For information about this choice, including consent withdrawal, please see our Privacy Policy.

Reviewer #1: No

---

## [Decision Letter · Decision Letter 1]

30 Nov 2023

PGPH-D-23-01290R1

Factors Associated with Overweight and Obesity among Women of Reproductive Age in Cambodia: Analysis of Cambodia Demographic and Health Survey

Dear Dr. Um,

Thank you for submitting your manuscript to PLOS Global Public Health. After careful consideration, we feel that it has merit but does not fully meet PLOS Global Public Health’s publication criteria as it currently stands. Therefore, we invite you to submit a revised version of the manuscript that addresses the points raised during the review process.

Reviewer 2 corresponded with me to clarify that the manuscript isn’t “wrong” just there are aspects that are debatable or not comprehensive enough. Can you revise in response to their comments, and focus on more cautious interpretation of odds ratios, to make sure you are not claiming to prove a causal link with observational data?

We look forward to receiving your revised manuscript.

Kind regards,

Abraham D. Flaxman, Ph.D.

Academic Editor

Journal Requirements:

Reviewers' comments:

Reviewer's Responses to Questions

**Comments to the Author**

1. If the authors have adequately addressed your comments raised in a previous round of review and you feel that this manuscript is now acceptable for publication, you may indicate that here to bypass the “Comments to the Author” section, enter your conflict of interest statement in the “Confidential to Editor” section, and submit your "Accept" recommendation.

Reviewer #2: (No Response)

2. Does this manuscript meet PLOS Global Public Health’s publication criteria? Is the manuscript technically sound, and do the data support the conclusions? The manuscript must describe methodologically and ethically rigorous research with conclusions that are appropriately drawn based on the data presented.

Reviewer #2: Partly

3. Has the statistical analysis been performed appropriately and rigorously?

Reviewer #2: No

4. Have the authors made all data underlying the findings in their manuscript fully available (please refer to the Data Availability Statement at the start of the manuscript PDF file)?

Reviewer #2: Yes

5. Is the manuscript presented in an intelligible fashion and written in standard English?

Reviewer #2: Yes

6. Review Comments to the Author

Reviewer #2: This manuscript investigated the factors associated with overweight and obesity in Cambodia using the latest Demographic and Health Survey (DHS) data. It focused on women of reproductive age and examined sociodemographic variables and selected behavioral risk factors such as use of contraceptives, smoking, and alcohol use. While the study provided some insight into the obesity epidemic in Cambodia, several aspects could be improved:

1. Factors associated with overweight and obesity: The authors acknowledged the omission of diet and physical activity from the study's considered factors, which are crucial direct influences on overweight and obesity outcomes. Although understanding the contribution sociodemographic factors to overweight and obesity is useful, it only offers a distal perspective, leaving a gap in understanding the most immediate factors.

2. Goodness of fit: The manuscript does not clarify the extent to which the selected variables in the multiple logistic regression account for variability in the data. Questions remain regarding the fit of the regression model and the proportion of unexplained variance.

3. Interpretation of AOR: Setting aside concerns of multicollinearity and model fit, the interpretation of results in Table 3 is convoluted. The odds ratios of several factors shifted from significant to non-significant after adjustment, with urban versus rural residency as an example. This change was likely due to the inclusion of other relevant factors such as occupation and wealth. However, the change of significance did not imply occupation being more important than place of residence. It was unclear, what conclusion should be drawn from the results in Table 3 comparing between OR and AOR.

4. Comparisons with the 2014 study: The study could benefit from a more detailed comparison with the DHS 2014 study. The current manuscript essentially replicated the previous study with new DHS data and the conclusion were similar. It would be informative to delineate the differences, changes over the decade, and any new insights provided by the latest research.

Other comments:

- Page 4, line 120: The term "height and length" should be revised to "height" when referring to adults.

- The general flow and coherence between sentences and paragraphs could be enhanced with editorial support.

7. PLOS authors have the option to publish the peer review history of their article (what does this mean?). If published, this will include your full peer review and any attached files.

**Do you want your identity to be public for this peer review?** For information about this choice, including consent withdrawal, please see our Privacy Policy.

Reviewer #2: No

---

## [Decision Letter · Decision Letter 2]

9 Jan 2024

Factors Associated with Overweight and Obesity among Women of Reproductive Age in Cambodia: Analysis of Cambodia Demographic and Health Survey

PGPH-D-23-01290R2

Dear Mr. Um,

We are pleased to inform you that your manuscript 'Factors Associated with Overweight and Obesity among Women of Reproductive Age in Cambodia: Analysis of Cambodia Demographic and Health Survey' has been provisionally accepted for publication in PLOS Global Public Health.

Best regards,

Abraham D. Flaxman, Ph.D.

Academic Editor

Reviewer Comments (if any, and for reference):

Reviewer's Responses to Questions

**Comments to the Author**

1. If the authors have adequately addressed your comments raised in a previous round of review and you feel that this manuscript is now acceptable for publication, you may indicate that here to bypass the “Comments to the Author” section, enter your conflict of interest statement in the “Confidential to Editor” section, and submit your "Accept" recommendation.

Reviewer #2: All comments have been addressed

2. Does this manuscript meet PLOS Global Public Health’s publication criteria? Is the manuscript technically sound, and do the data support the conclusions? The manuscript must describe methodologically and ethically rigorous research with conclusions that are appropriately drawn based on the data presented.

Reviewer #2: Yes

3. Has the statistical analysis been performed appropriately and rigorously?

Reviewer #2: Yes

4. Have the authors made all data underlying the findings in their manuscript fully available (please refer to the Data Availability Statement at the start of the manuscript PDF file)?

Reviewer #2: Yes

5. Is the manuscript presented in an intelligible fashion and written in standard English?

Reviewer #2: Yes

6. Review Comments to the Author

Reviewer #2: (No Response)

7. PLOS authors have the option to publish the peer review history of their article (what does this mean?). If published, this will include your full peer review and any attached files.

**Do you want your identity to be public for this peer review?** For information about this choice, including consent withdrawal, please see our Privacy Policy.

Reviewer #2: No
